# META-LEARNING FOR BATCH MODE
# ACTIVE LEARNING

**Sachin Ravi**
Department of Computer Science
Princeton University, NJ, USA
sachinr@princeton.edu

**Hugo Larochelle**
Google Brain
Montreal, CA
hugolarochelle@google.com

## ABSTRACT

Active learning involves selecting unlabeled data items to label in order to best improve an existing classifier. In most applications, *batch mode active learning*, where a set of items is picked all at once to be labeled and then used to re-train the classifier, is most feasible because it does not require the model to be re-trained after each individual selection and makes most efficient use of human labor for annotation. In this work, we explore using meta-learning to learn an active learning algorithm that selects the best set of unlabeled items to label given a classifier trained on a small training set. Our experiments show that our learned active learning algorithm is able to construct labeled sets that improve a classifier better than commonly used heuristics.

## 1 INTRODUCTION

Active learning methods aim to use the existing classifier in some way so as to decide which unlabeled items are best to label in order to improve the existing classifier the most. The majority of popular approaches are based on heuristics such as choosing the item whose label the model is most uncertain about, choosing the item whose addition will cause the model to be least uncertain about other items, or choosing the item that is most "different" compared to other unlabeled items according to some similarity function (Joshi et al., 2009; Houlsby et al., 2011; Sener & Savarese, 2017). Though these heuristics work well, they are motivated in the context where instances to label are selected one at a time, re-training the model at every step. On the other hand, it is often more appropriate and efficient to send data for labeling in batch mode, i.e. requesting that a *set* of instances be labeled by people. The heuristics mentioned above can be extended to the batch setting by taking the $B$ best items according to the heuristic's metric of selection; however, this can lead to substantially suboptimal performance and produce sets with overly redundant items. Though there has been work on algorithms specifically for batch mode active learning (Hoi et al., 2006; Guo & Schuurmans, 2008), these methods involve a complexity for selecting each new item that is at least quadratic in the unlabeled set size, making them prohibitive to use for large unlabeled sets. Furthermore, most work on active learning has assumed that the unlabeled items belong to at least one of the classes we are interested in classifying; however, real data does not obey these constraints and unlabeled data will often consist of *distractor* items (Ren et al.) that do not belong to any of the classes we are considering. These distractor items pose a problem because they may be items that are viewed favorably by the above heuristics, but they should not be labelled as they will not help the classifier. We are thus interested in the following desiderata for a batch active learning method: 1) it should be designed to directly optimize its effectiveness at finding sets of items to label that improve the model's ultimate performance; 2) it should have linear complexity in selecting each additional item; 3) it should work well in the presence of distractors. Previous work has explored meta-learning for active learning but it has not been applied in the batch mode setting or with distractors (see appendix).

Active learning is most effective and valuable in situations where the initial amount of labeled data is small. Hence, we formulate the task of batch mode active learning under the few-shot learning framework, so as to learn a meta-learning model optimized for this problem. To do so, we build on Prototypical Networks (Snell et al., 2017), a state-of-the-art approach to few-shot learning and augment it with a active learning model that can select a subset of new examples for labeling to improve its classification accuracy. We demonstrate empirically that our learned algorithm for batch

mode active learning constructs better sets for labeling when compared to commonly used heuristics.

## 2 META-LEARNING FOR BATCH MODE ACTIVE LEARNING

We can modify the episode structure mentioned in Snell et al. (2017) so as to accommodate a batch mode active learning objective by supplementing the support and query sets with an unlabeled set $\mathcal{U} = \{\tilde{\mathbf{x}}_1, \ldots, \tilde{\mathbf{x}}_M\}$ that consists of $M$ unlabeled examples. We consider $K$-shot, $N$-class, $B$-batch episodes where we need to select a subset $\mathcal{A} \subseteq \mathcal{U}$ of size $B$ to be labeled and added to our support set $\mathcal{S}$ to get a new support set $\mathcal{S}' = \mathcal{S} \cup \mathcal{A}$. The goal is to use the classifier formed from the original support set $\mathcal{S}$ to select the best subset of $B$ examples from $\mathcal{U}$ to label to create the new support set $\mathcal{S}'$ and associated new classifier so as to most improve the performance on the query set $\mathcal{Q}$.

We represent the probability of a $B$-sized subset $\mathcal{A} = \{\tilde{x}_1, \ldots, \tilde{x}_B\} \subseteq \mathcal{U}$ by decomposing the probability using the chain rule: $p(\mathcal{A}) = p(\tilde{x}_1, \ldots, \tilde{x}_B) = \prod_{i=1}^{B} p(\tilde{x}_i \mid \tilde{x}_1, \ldots \tilde{x}_{i-1})$. Assuming, we have already built up a subset $\mathcal{A} = \{\tilde{x}_1, \ldots, \tilde{x}_j\}$ for $j < B$, we show how we parameterize the distribution $p(\tilde{x} \mid \mathcal{A}) = p(\tilde{x} \mid \tilde{x}_1, \ldots, \tilde{x}_j)$ to select the next unlabeled item to add to our subset $\mathcal{A}$. We keep sampling from this distribution and adding the sampled item to our subset until our subset $\mathcal{A}$ has $B$ items. We can represent our prototypical-net classifier attained from the original support set $\mathcal{S}$ by the set of prototypes $\{c_k\}_{k=1}^{K}$ computed for each of $K$ classes. We can calculate a set of statistics relating each unlabeled item $\tilde{x}_i \in \mathcal{U}$ to the set of prototypes and we denote this set of item-classifier statistics by $\Pi\left(\{c_k\}_{k=1}^{K}, \tilde{x}_i\right)$. We will use these statistics as input to compute two distributions $p_{quality}(\tilde{x})$ and $p_{diversity}(\tilde{x} \mid \tilde{x}_1, \ldots \tilde{x}_j)$, which represent two different probability distributions over which unlabeled item to add next to the existing subset $\mathcal{A}$.

**Quality Distribution** Using the set of statistics for each unlabeled item, we can compute the probability of selecting an unlabeled item according to its quality as

$$p_{quality}(\tilde{x}_i) \propto \exp(q_i), \text{ where } q_i = f_{\mathbf{q}}\left(\Pi\left(\{c_k\}_{k=1}^{K}, \tilde{x}_i\right)\right),$$

where $f_q$ is a MLP with parameters $q$. This distribution independently maps the probability of each unlabeled item being selected based on a prediction of how useful the item will be to the existing classifier according to a learned function of item-classifier statistics.

**Diversity Distribution** The same set of statistics can also be used to compute a feature vector describing the unlabeled item to classifier relationship as

$$\phi_i = f_{\phi}\left(\Pi\left(\{c_k\}_{k=1}^{K}, \tilde{x}_i\right)\right),$$

where $\phi_i \in \mathbb{R}^{D'}$ and $f_{\phi}$ is a MLP with parameters $\phi$. The goal of the diversity distribution is to increase the probability of selecting unlabeled items which are dissimilar from the items that already make up the set $\mathcal{A} = \{\tilde{x}_1, \ldots, \tilde{x}_j\}$ where similarity is measured in terms of each item's corresponding feature vector. The probability of selecting an unlabeled item according to its diversity is then:

$$p_{diversity}(\tilde{x}_i \mid \mathcal{A}) \propto \exp(v(\phi_i)/\tau), \text{ where } v(\phi_i) = \min_{\tilde{x}_j \in \mathcal{A}}\{\sin\theta_{ij}\},$$

where $\theta_{ij}$ is the angle between feature vectors $\phi_i$ and $\phi_j$ and $\tau$ is a learned temperature parameter that allows us to control the flatness of this distribution. The probability of an item being picked increases as its feature vector is more orthogonal to feature vectors corresponding to items already having been added to the subset $\mathcal{A}$.

**Product of Experts** The final probability distribution over which unlabeled item to add to the subset $p(\tilde{x} \mid \mathcal{A})$ is attained as a product of experts model combining the distributions $p_{quality}$ and $p_{diversity}$:

$$p(\tilde{x} \mid \mathcal{A}) \propto p_{quality}(\tilde{x}) \cdot p_{diversity}(\tilde{x} \mid \mathcal{A}) \cdot \mathbb{1}_{\tilde{x} \notin \mathcal{A}},$$

where the indicator variable enforces not having any support over an item that already belongs to $\mathcal{A}$.

**Training** We want to update the parameters $\theta' = \{\phi, \mathbf{q}, \tau\}$ of our model so that for any given episode with corresponding support set $\mathcal{S}$, unlabeled set $\mathcal{U}$, and query set $\mathcal{Q}$, $p_{\theta'}(\tilde{x} \mid \mathcal{A})$ can be repeatedly sampled to create a subset $\mathcal{A} \subseteq \mathcal{U}$ of size $B$ so that the prototypes computed on the new

| Method | $B = 5$ $M = 100$ | $B = 10$ $M = 100$ | $B = 20$ $M = 200$ | $B = 5$ w/ D $M = 200$ | $B = 10$ w/ D $M = 200$ | $B = 20$ w/ D $M = 400$ |
|---|---|---|---|---|---|---|
| Max-Entropy | -0.5% | +5.0% | +11.3% | -3.3% | -1.3% | +2.8% |
| Min-Max Sim | +0.3% | +3.6% | +7.7% | -0.8% | -0.1% | +2.0% |
| Random | +1.6% | +8.6% | +17.8% | -2.9% | +2.1% | +7.5% |
| Meta-Learner | **+6.6%** | **+11.1%** | **+18.1%** | **+1.3%** | **+6.1%** | **+11.20%** |

Table 1: results for *mini*ImageNet. The first set of columns indicate the non-distractor setting and the second set of columns indicate the distractor setting (w/ D). Percentages are improvement in accuracy on the query set over initial prototypical net averaged over 1000 episodes. $B$ indicates the size of the subset we are allowed to label and $M$ indicates the size of the unlabeled set in each episode.

support set $\mathcal{S}' = \mathcal{S} \cup \mathcal{A}$ have high classification performance on the query set $\mathcal{Q}$. We train our model on each episode via the REINFORCE gradient with a leave-one-out baseline (Mnih & Rezende, 2016). The gradient we use is the following:

$$\nabla_{\theta'} \mathbb{E}_{p_{\theta'}(\mathcal{A})}[\mathcal{C}(\mathcal{Q} \,|\, \mathcal{S} \cup \mathcal{A})] \approx \frac{1}{T} \sum_{t=1}^{T} [(\mathcal{C}(\mathcal{Q} \,|\, \mathcal{S} \cup \mathcal{A}_t) - \beta_{-t}) \nabla_{\theta'} \log p_{\theta'}(\mathcal{A}_t)], \text{ where } \mathcal{A}_t \sim p_{\theta'}(\mathcal{A}),$$

where $\mathcal{C}(\mathcal{Q} \,|\, \mathcal{S} \cup \mathcal{A})$ is the accuracy of the prototypical net on the query set $\mathcal{Q}$ when the support set is $\mathcal{S}' = \mathcal{S} \cup \mathcal{A}$ and $\beta_{-t} = \frac{1}{T-1} \sum_{t' \neq t}^{T} \mathcal{C}_\theta(\mathcal{Q} \,|\, \mathcal{S} \cup \mathcal{A}_{t'})$ is the baseline. Additionally, we can supplement the reward of the classification accuracy of the query set $\mathcal{Q}$ with the classification accuracy of all items in the unlabeled set $\mathcal{U}$, as during training we have labels for these items. This reward gives the model the correct feedback that the items it picks to be labeled should help the classifier classify all other items in the unlabeled set. Thus, rather than using $\mathcal{C}(\mathcal{Q} \,|\, \mathcal{S} \cup \mathcal{A})$, we use $\mathcal{C}(\mathcal{Q} \cup \mathcal{U} \,|\, \mathcal{S} \cup \mathcal{A})$.

## 3 EXPERIMENTS

We evaluate the performance of our learned batch active-learning algorithm on 2 different few-shot learning benchmarks: **CIFAR-100** and *mini***ImageNet**, where both datasets consist of 100 classes and 600 images per class and where CIFAR-100 has images of size $32 \times 32$ and *mini*ImageNet has images of size $84 \times 84$. We split the 100 classes into separate sets of 64 classes for training, 16 classes for validation, and 20 classes for testing for both of the datasets. For our initial classifier, we use a prototypical net trained on 1-shot, 5-class episodes for each of the datasets and we evaluate adding unlabeled items to a 1-shot support set with various different batches $B$.

We compare against 3 commonly used heuristics for active-learning: (1) *Max-Entropy*: pick the item whose classification probability according to the original classifier has the highest entropy; (2) *Min-Max Sim*: pick the item which has the the smallest maximal similarity to other unlabeled items; (3) *Random*: randomly pick an unlabeled item from the set. We consider adding items in both the non-distractor and distractor setting, where the unlabeled set either does not contain distractor items or does contain them, respectively. For each different batch $B$ considered, there is an associated size of the unlabeled set $M$, which is generated to have equal amount of examples from the 5 classes being considered in each episode. In the distractor setting, we assume there are an equal number of 5 distractor classes and generate an equal number of distractor unlabeled examples from these classes to supplement the unlabeled set. If a distractor item is picked to be labeled, we do not add it the support set, indicating the lack of benefit of these items. We evaluate all methods on the same fixed number of 1000 sampled episodes and so do not consider confidence intervals. Results comparing the heuristics against the learned meta-learner for *mini*ImageNet are shown in Table 1, with CIFAR-100 results given in appendix. We can see that the learned batch mode active learning strategy outperforms the heuristics on all of the different cases considered.

## 4 CONCLUSION & FUTURE WORK

We propose a meta-learning model for batch mode active learning that improves the accuracy of a classifier better than commonly used heuristics on two few-shot learning benchmarks. Future work involves evaluating the method to see how it scales with a larger support set and understanding how the trade off between quality and diversity is occurring in selection of unlabeled items.

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

## A  EXPERIMENT DETAILS

### A.1  RESULTS FOR CIFAR-100

Results comparing the heuristics against the learned meta-learner for CIFAR-100 are shown in Table 2. These results follow the same trend as before, as the learned active-learning algorithm outperforms the heuristics in all cases.

| Method | $B = 5$ $M = 100$ | $B = 10$ $M = 100$ | $B = 20$ $M = 200$ | $B = 5$ w/ D $M = 200$ | $B = 10$ w/ D $M = 200$ | $B = 20$ w/ D $M = 400$ |
|---|---|---|---|---|---|---|
| Max-Entropy | +1.5% | +7.3% | +13.8% | -1.4% | +0.7% | +5.3% |
| Min-Max Sim | +2.2% | +5.3% | +10.0% | +0.9% | +2.1% | +4.5% |
| Random | +3.3% | +10.0% | +17.2% | -1.1% | +2.6% | +8.6% |
| Meta-Learner | **+7.3%** | **+11.6%** | **+17.4%** | **+3.2%** | **+6.2%** | **+10.80%** |

Table 2: results for CIFAR-100. The first set of columns indicate the non-distractor setting and the second set of columns indicate the distractor setting (w/ D). Percentages are improvement in accuracy on the query set over initial prototypical net averaged over 1000 episodes. $B$ indicates the size of the subset we are allowed to label and $M$ indicates the size of the unlabeled set in each episode.

## A.2 MODEL ARCHITECTURE

Our prototypical network uses the original CNN architecture defined in Vinyals et al. (2016). We calculate $f_\phi$ as a single-layer that embeds the item-classifier statistics into a 40-dimensional embedding. The MLP used for $f_q$ is a 3-layer net with 40 hidden units each, where the final layer projects into the quality logit. We use batch-normalization across the unlabeled set to normalize the hidden units.

## A.3 ITEM-CLASSIFIER STATISTICS

The item-classifier statistics we use are the following:

- Distance Statistics: these consists of the square euclidean distance from the unlabeled item to each of the prototypes and statistics about these distances, including the min, the max, the mean, the variance, the skew, and the kurtosis.

- Classification Probability Statistics: these consist of the classification probability distribution for the unlabeled item according to the prototypes and statistics about this distribution, including the entropy, the min, the max, the variance, the skew, and the kurtosis.

## B RELATED WORK

Our proposed model is related to the Determinantal Point Process (DPP) (Kulesza & Taskar, 2011b;a), which maintains a probability distribution overall the set of all subsets using sub-determinants of a kernel matrix, which capture both the quality and diversity of items in the subset. A DPP gives higher probability to sets whose corresponding vectors span a larger volume parallelepiped (based on the geometric interpretation of the determinant as computing volume). For an individual item in the set, the probability of the set is increased if the length of the corresponding vector increases (the quality of the item increases) and/or the corresponding vector becomes less similar to the vectors corresponding to other items in the subset (the diversity of the subset increases). We attempt to capture both of these properties in our model while avoiding the heavy computational overhead of a DPP, as sampling relies on an eigendecomposition of the DPP kernel, which has cubic complexity in the size of the unlabeled set.

Additionally, our model bears similarities to the Neural Cache Model (NCM) (Grave et al., 2016), which adds a cache-like memory to neural network language models in order to adapt their predictions to recent history. Whereas the NCM uses its cache to select a hidden state (and associated vocabulary item) in the cache that is most similar to the current hidden state, we use the cache to select the most different feature vector (and associated unlabeled item) outside of the cache from the feature vectors stored in the cache. Furthermore, the NCM uses a mixture of experts model to combine predictions from the regular language model and the cache-based model whereas we use a product of experts model (Hinton, 2002; Welling, 2007) to combine predictions, as a product of experts encodes the AND relation we want between our quality and diversity distributions.

Though previous work has considered learning a model to perform data selection for labelling, it has been applied to the setting where the classifier is re-trained at each step and mainly been applied to smaller datasets (Konyushkova et al., 2017; Woodward & Finn, 2017; Fang et al., 2017). The

most relevant previous work to ours is by Bachman et al. (2017), who also consider meta-learning for learning an active learning algorithm. Their work, however, does not consider the batch mode active learning setup and involves a model that learns to build up a training set from scratch, with the classifier being recomputed after each selection step. Moreover, in the few-shot classification setting (as considered in this work), the performance of their model is on par with the *Min-Max-Sim* heuristic, which we consider and compare with in our experiments. Lastly, none of the previous work consider the more challenging but realistic scenario of the unlabeled set containing distractors.

