# OpenReview forum: "Meta-Learning for Batch Mode Active Learning"
_ICLR.cc/2018/Workshop — Accept_

### Official Review · AnonReviewer3 · 2018-03-08
**Probabilistic View of Batch Mode Active learning**

**Rating:** 6
**Confidence:** 3

**Review:**

The paper presents a probabilistic view of batch mode active learning. It will select a subset of unlabeled data to query by balancing the quality and diversity of the queried samples.

The paper introduces novel definition of quality distribution and diversity distribution, which is interesting. However, the paper is not easy to read. Some points should be clarified.

1. In active learning, the query set is the unlabeled samples to be queried for labeling. However, I am not sure if the "query set" in this paper is the same definition. It confuse me.

2. Most batch mode active learning is to balance the discrimination and diversity of the queried set. The general idea in this paper is not new. The authors are welcome to explain why they propose this new definition of quality distribution and diversity distribution. Why in current form?

3. The authors should compare their proposed algorithm with state-of-the-art batch mode active learning algorithms. The current baselines are far from convincing.

4. One limitation of the proposed algorithm is that it requires a multi-class classification problem with large number of classes. The authors are encouraged to clarify that if I am wrong.

---

### Official Review · AnonReviewer2 · 2018-03-09
**Good paper with promising results.**

**Rating:** 8
**Confidence:** 4

**Review:**

This work proposes a meta-learning method for batch model active learning. Different from existing models, it adds a diversity distribution in the selecting criteria to represent the mutual effects among selected items. The final performance of the method seems promising.
I like the paper but have following questions:
1.    It’s not clear which part of the model considers distractor explicitly.
2.    Do you test the influence of the quality distribution and the diversity distribution separately?
3.    How does the training speed of the REINFORCE on this setting like? Since the training procedure seems not very easy.
4.    Some related batch-mode active learning papers are missing, such as:
[1] Gu Q, Zhang T, Han J. Batch-mode active learning via error bound minimization[C]//UAI.[S.l.: s.n.], 2014.
[2] Chaudhuri K, Kakade S M, Netrapalli P, et al. Convergence rates of active learning for maximum likelihood estimation[C]//NIPS. [S.l.: s.n.], 2015.
5.    Some typos need to be fixed.

---

### Official Review · AnonReviewer1 · 2018-03-11
**Novel problem and approach, a well-written paper, experiments have space for improvement.**

**Rating:** 6
**Confidence:** 4

**Review:**

Overall, this paper is well-written. The authors give a thorough and detailed review of the main difference between classic active learning (AL) and batch active learning. The motivation of using batch AL is intuitive and the proposed framework has a clear structure which considers both the query item's quality and query item's diversity.

However, a few things could be improved, and below are the detailed comments:
- The authors simply utilize the product of the probability of quality and diversity as the criteria. What is the intuitive of using such kind of final distribution? Are these two have to be independent? It would be interesting to consider their correlation as well. With the increasing number of items, the diversity of items (defined using the vector angle) could be very difficult to be guaranteed. This diversity criterion could have limited selection power in your batch AL.
- Regarding the experiment, the selection of parameters B and M seems like to be heuristic choices. Please elaborate the reasons for using these values.
- The experiment only considers the batch AL. If the authors have claimed the effectiveness of batch AL, the comparison between batch AL and AL should be listed in the experimental section.

---

### Decision · Program_Chairs · 2018-03-20
**ICLR 2018 Workshop Acceptance Decision**

**Decision:**

Accept

**Comment:**

Congratulations, your paper was accepted to the ICLR workshop.